# Changes in Plasma Glial Fibrillary Acidic Protein in Children Receiving Sevoflurane Anesthesia: A Preliminary Randomized Trial

**DOI:** 10.3390/jcm10040662

**Published:** 2021-02-09

**Authors:** Eun-Hee Kim, Young-Eun Jang, Sang-Hwan Ji, Ji-Hyun Lee, Sung-Ae Cho, Jin-Tae Kim, Hyunyee Yoon, Hee-Soo Kim

**Affiliations:** 1Department of Anesthesiology and Pain Medicine, Seoul National University Hospital, Seoul National University College of Medicine, 101 Daehak-ro, Jongno-gu, Seoul 03080, Korea; beloveun@gmail.com (E.-H.K.); na0ag2@hotmail.com (Y.-E.J.); taepoongshin@gmail.com (S.-H.J.); muslab@hanmail.net (J.-H.L.); whtjddo89@gmail.com (S.-A.C.); jintae73@gmail.com (J.-T.K.); 2Protein Immunology Core Facility, Biomedical Research Institute, Seoul National University Hospital, Seoul 03082, Korea; hyunyee@snuh.org

**Keywords:** dexmedetomidine, general anesthesia, glial fibrillary acidic protein, pediatrics

## Abstract

We investigated changes in plasma glial fibrillary acidic protein concentration during sevoflurane anesthesia induction in children < 3 years old and determined the effect of co-administering dexmedetomidine. This preliminary randomized trial included 60 pediatric patients who received sevoflurane anesthesia for >3 h. Patients were assigned to dexmedetomidine or control groups at a 1:1 ratio. The primary outcome was changes in plasma glial fibrillary acidic protein concentration of dexmedetomidine and control groups over time. Fifty-five patients were included in the final analysis. The median (interquartile range (IQR)) of the plasma glial fibrillary acidic protein level was 387.7 (298.9–510.8) pg·mL^−1^ immediately after anesthetic induction, 302.6 (250.9–412.5) pg·mL^−1^ at 30 min, and 321.9 (233.8–576.2) pg·mL^−1^ at 180 min after the first sample. These values did not change over time (*p* = 0.759). However, plasma glial fibrillary acidic protein increased after 180 min of infusion of dexmedetomidine compared with values at 30 min infusion (*p* = 0.04, mean difference and 95% confidence interval of 221.6 and 2.2 to 441.0 pg·mL^−1^). In conclusion, three hours of sevoflurane anesthesia in pediatric patients < 3 years old did not provoke neuronal injury assessed by the plasma biomarker. Further studies regarding the effect of prolonged dexmedetomidine infusion on anesthetic neuronal injury are required.

## 1. Introduction

Annually, more than a million pediatric patients receive anesthetic agents for general anesthesia or sedation, including 1 in 7 children who are exposed to anesthetic agents before an age of 3 years [1,2]. The growing demand for surgery and anesthesia is accompanied by a consequential need for safe anesthetic strategies [3].

The search for anesthetic management with attenuated neuronal injury is timely and highly warranted, and various preclinical studies have indicated the promising activity of both dexmedetomidine and xenon [4]. However, there is a dearth of clinical studies in actual pediatric patients.

Glial fibrillary acidic protein (GFAP) and ubiquitin C-terminal hydrolase-L1 are the first FDA-approved blood biomarkers for the diagnosis of traumatic brain injury [5,6]. Furthermore, the increase in plasma GFAP concentration and neurologic outcomes are highly relevant in patients with traumatic brain injuries, those undergoing surgery with cardiopulmonary bypass, and survivors of cardiac arrest [7,8,9,10,11,12,13]. Recently, several studies attempted to identify the association between the perioperative management and plasma GFAP concentrations [14,15,16]. However, research on pediatric surgical patients has not yet been performed.

In this preliminary randomized study, we aimed to determine changes in plasma GFAP concentrations associated with prolonged sevoflurane anesthesia in children < 3 years old and the potential effects of co-administration of dexmedetomidine on these changes.

## 2. Materials and Methods

### 2.1. Trial Design and Ethics

This preliminary study was approved by the institutional review board (IRB) of Seoul National University Hospital (IRB no. 1706-132-861), and written informed consent was obtained from the guardians of the pediatric patients participating in the trial. The trial was registered prior to patient enrollment at the clinicaltrials.gov protocol registration and results system (NCT03234660, Principal investigator: Professor Hee-Soo Kim, Date of registration 17 July 2017). We conducted an open label, randomized controlled trial where patients were allocated at a 1:1 ratio to the dexmedetomidine and control groups. This study was conducted in strict compliance with the ethics of the declaration of Helsinki 2013.

### 2.2. Participants

This study was conducted at a single tertiary care children’s hospital in Seoul, Korea. Pediatric patients < 3 years old who were scheduled to receive general anesthesia for >3 h were included. Exclusion criteria were as follows: previous history of general anesthesia, preoperatively diagnosed neurocognitive dysfunction, cardiopulmonary bypass requirement, anticipated hemodynamic instability during surgery (massive bleeding and transfusion), and significantly elevated liver enzyme profiles (aspartate transaminase and alanine aminotransferase > 100 unit·L^−1^). 

### 2.3. Interventions

In the operating theater, patients were fitted for electrocardiogram (ECG), non-invasive blood pressure, pulse oximetry, and bispectral index measurements. Anesthesia was induced using 0.02 mg·kg^−1^ atropine, 5 mg·kg^−1^ thiopental sodium, 4–8 vol% sevoflurane, and 0.1 μg·kg^−1^·min^−1^ remifentanil. Tracheal intubation was facilitated by 0.6 mg·kg^−1^ rocuronium for muscle relaxation. Immediately after invasive arterial catheter placement, blood samples were collected for baseline GFAP plasma concentration measurement (baseline, before dexmedetomidine infusion, T1). Then, each patient received the allocated intervention throughout the surgery.

The dexmedetomidine group received 1 μg·kg^−1^ dexmedetomidine for 10 min followed by 0.5 μg·kg^−1^·h^−1^ until the end of the operation. Remifentanil was continued during the surgery at a rate of 0.1 μg·kg^−1^·min^−1^. Sevoflurane was maintained as guided by a bispectral index (BIS) of 40–70 during the surgery. In the control group, sevoflurane was maintained at 2.5–3.5 vol% guided by a BIS of 40–70, while remifentanil was continued at a rate of 0.1 μg·kg^−1^·min^−1^ during the surgery. When the systolic blood pressure was <60, 70, and 75 mmHg for neonates, infants, and those 1–3 years old, respectively, continuous infusion of 5–7 μg·kg^−1^·min^−1^ dopamine was initiated. Because the intraoperative vasoactive reactions expressed by heart rate and blood pressure variations are dependent on the depth of anesthesia, sevoflurane was maintained guided by a BIS of 40–70 during the surgery with a target value of around 60 in order to enable the further precision of remifentanil dosing. Additionally, as BIS happens to show aberrant values during different stages of general anesthesia using sevoflurane and rocuronium, the BIS values were on-line verified against observance of fraction of inspired anesthetic agent, fraction of expired anesthetic agent, and mean alveolar concentration [17,18,19,20,21].

### 2.4. Blood Sampling and Analysis

Blood samples for GFAP analysis were collected three times per patients. Two milliliters of arterial blood was collected into tubes containing ethylenediaminetetraacetic acid (EDTA) immediately after induction (T1) and 30 (T2) and 180 (T3) min after the first sampling. All samples were centrifuged for 10 min at 3000 rpm and immediately stored at −80 °C. They were thawed immediately before assay and diluted 2-fold with assay diluent (Meso Scale Diagnostics, Cat. No. R51AD-3). Quality control (QC) samples of low (2 ng·mL^−1^), medium (10 ng·mL^−1^), and high GFAP (50 ng·mL^−1^) levels were freshly prepared by spiking of the human GFAP calibrator (Meso Scale Diagnostics, Cat. No. C001M-2) in assay diluent. Stock solution of the human GFAP calibrator was diluted 20-fold with assay diluent to the highest standard. Then, calibrator samples were freshly prepared for the 7-point calibration curve by 4-fold dilution with assay diluent. The expression levels of GFAP were measured by Meso Scale Diagnostics (MSD) multi-array technology using highly sensitive electrochemiluminescence (ECL) detection following the manufacturer’s instructions. The MSD R-PLEX human GFAP antibody set (Meso Scale Diagnostics, Cat. No. F211M-3, Rockville, MD, USA) and MSD GOLD 96-well small spot streptavidin plates (Meso Scale Diagnostics, Cat. No. L45SA-2) were used in the assays. The assay plates for the target protein were detected the ECL signals by a Meso Sector S 600 (Meso Scale Diagnostics, MD 20850 USA) with Discovery Workbench Software (version 4.0.12.1). All assays were conducted in duplicate within a batch and also qualified the GFAP levels (pg·mL^−1^) of various samples, including the plasmas, QCs, and calibrators. The calibration curves were evaluated in six independent batches, and the QC samples were analyzed in every batch. The means of the lower limit of detection (LLOD) and ultra-limit of detection (ULOQ) for the calibration curves were 128.5 pg·mL^−1^ and 504.6 ng·mL^−1^, respectively. All calibrators and QC samples were satisfied with the acceptance criteria, including 20 percent of the coefficient of variations (CVs) in inter-batch assays.

### 2.5. Outcomes

The primary outcome was time-related changes in plasma GFAP concentration of the dexmedetomidine and control groups. Secondary outcomes were mean sevoflurane concentration, mean bispectral index, and the incidence of arrhythmia and systolic hypotension (systolic hypotension was defined by the pediatric advanced life support algorithm 2019 and systolic blood pressure < 60, 70, and 75 mmHg for neonates, infants, and those 1–3 years old, respectively).

### 2.6. Sample Size Calculation

Because this study was a preliminary pilot study and there were no historical data available on the changes of GFA in pediatric patients with sevoflurane anesthesia with or without dexmedetomidine, no sample size calculation was performed. Therefore, we aimed at a sample size of 30 patients in each group.

### 2.7. Randomization and Blinding

To establish the study groups, an independent researcher who was not involved with the study details managed and concealed the randomization process and assigned the pediatric patients to each group. A research assistant nurse prepared 1 μg·mL^−1^ dexmedetomidine hydrochloride (Precedex, 100 μg·mL^−1^, Pfizer Canada Inc., Kirkland, QC, Canada) in a 50 mL syringe. 

The group assignment was not blinded to the attending anesthesiologist who controlled the sevoflurane concentration. We tried to minimize the sevoflurane concentration in patients receiving dexmedetomidine. Furthermore, only the outcome assessor (S.-H. Ji) was blinded to the group assignment.

### 2.8. Statistical Methods

The Kolmogorov–Smirnov test was used to evaluate the normality of the data, while the chi-squared test was used to compare categorical data. Student’s *t*-test, the Mann–Whitney U test, and a paired *t*-test were used to compare continuous data. One-way ANOVA was performed to compare GFAP concentration over time in all patients. A mixed-effects analysis of variance (ANOVA) was performed to compare GFAP plasma concentrations over time between the control and dexmedetomidine groups. For all analyses, a *p* < 0.05 was considered statistically significant. MedCalc version 18.11.3 (MedCalc, Mariakerke, Belgium) and statistical package for the social sciences (SPSS) version 23.0 (IBM Corporation, Armonk, NY, USA) were used for the statistical analysis.

## 3. Results

Sixty pediatric patients were assessed for eligibility. After excluding two who did not meet the inclusion criteria (previous history of general anesthesia and abnormal liver enzyme profile) and one who declined to participate, 29 and 28 patients were allocated to the control and dexmedetomidine groups, respectively. Furthermore, the allocated intervention was discontinued in two patients included in the dexmedetomidine group because of intravenous catheter malfunction. Finally, 55 patients were included in the final analysis (Figure 1).

The first and last participant were registered on 19 September 2017 and 4 September 2019, respectively. Table 1 shows the baseline demographic and clinical characteristics of the two groups, which did not differ in baseline demographics, total anesthesia time, operation type, bispectral index, and duration of hypotension. However, the end tidal sevoflurane concentration was lower in the dexmedetomidine group than it was in the control group with a mean difference (95% confidence interval (C)) of −1.25 (−1.53 to −0.96).

Table 2 and Figure 2 show plasma GFAP concentration during surgery. Plasma GFAP concentration did not change over time in all patients (*p* = 0.072). Furthermore, plasma GFAP concentration did not change in control group over time. However, that of the dexmedetomidine group increased from T2 to T3 (*p* = 0.047, mean difference and 95% confidence interval of 221.6 and 2.2 to 441.0 pg·mL^−1^). However, plasma GFAP concentration did not differ between T1 and T3 (*p* = 0.473). The mixed-effects ANOVA showed that plasma GFAP concentration changed over time (*p* = 0.04, F = 3.317), and the changes were different between the groups (*p* = 0.02, F = 5.48).

## 4. Discussion

We found no change relative to time in plasma GFAP concentration following sevoflurane anesthesia in pediatric patients < 3 years old. However, prolonged dexmedetomidine co-administration increased plasma GFAP concentration at 3 h of sevoflurane anesthesia.

We assumed that if the sevoflurane anesthesia provoked neuronal injury in pediatric surgical patients, plasma GFAP concentration would increase over time during sevoflurane anesthesia induction. However, plasma GFAP concentration did not increase during the induction of sevoflurane anesthesia. In line with this, Stacie et al. reported that inhalation anesthesia without surgery did not increase plasma GFAP in adults [16]. Our patients showed considerably high plasma GFAP concentrations during sevoflurane anesthesia induction (391.2 pg·mL^−1^ mean in all patients). The ALERT-TBI study demonstrated that patients with mild traumatic brain injury with a Glasgow coma scale (GCS) score of 9–15 showed a median (interquartile range (IQR)) plasma GFAP concentration of 24.3 (10.0–57.4) pg·mL^−1^ [22]. Douglas et al. reported that plasma GFAP concentration was markedly elevated to 600 ± 200 pg·mL^−1^ in pediatric patients with severe traumatic brain injury (2–17 years old; GCS score, 4.8 ± 0.1) [23]. However, considering a normal plasma GFAP level in normal pediatric controls with no known neurologic injury (median of 55 pg·mL^−1^, IQR: 0–92 pg·mL^−1^, 95th percentile of 436 pg·mL^−1^, similar across age [24]) and maintaining the level during the prolonged induction of sevoflurane anesthesia, we concluded that these values did not represent neuronal injury in our patients. It is difficult to make interpretations because we did not obtain the preoperative plasma GFAP value of these patients. We consider this as a study limitation, and further studies need to identify the effect of anesthetic induction on plasma GFAP concentration.

In our study, co-administration of dexmedetomidine after anesthetic induction could have increased plasma GFAP concentration after 180 min of anesthesia. Preclinical evidence has indicated that dexmedetomidine post-conditioning improves neurological outcomes after brain hypoxic–ischemic injury in neonatal rats [25] and provides neuroprotection against induced subarachnoid hemorrhage in rats, traumatic brain insult in mice, and glucose and oxygen deprivation in rats [26,27,28]. Sanders et al. reported that dexmedetomidine did not induce neuroapoptosis and that its co-administration (25 μg kg^−1^) decreased isoflurane-induced neurotoxicity by inhibiting caspase-3 expression and attenuating cortical apoptosis [29]. However, currently, there is limited clinical evidence of the neuroprotective effects of dexmedetomidine in children following neurological injury. Sanders et al. also emphasized that increasing the dexmedetomidine dose (150 μg kg^−1^) induced neuroapoptosis of cells in primary sensory brain regions [30]. Perez-zoghbi et al. also noted that dexmedetomidine co-administration at a high dose (5 μg kg^−1^) increased mortality of neonatal rats [31].

While this is a preliminary study, our results suggest that prolonged dexmedetomidine infusion may provoke neuronal injury, thereby warranting further studies to investigate the optimal dexmedetomidine dose and duration.

There are some limitations to our preliminary study. As abovementioned, we could not measure the preoperative level of plasma GFAP, which might provide important implications in terms of result interpretation. Moreover, as our sample size was considerably small, random confounding errors may have occurred. Large-scale studies are warranted to elucidate the effect of dexmedetomidine on plasma GFAP concentration during induction of sevoflurane anesthesia. Lastly, two patients in the dexmedetomidine group showed notably different plasma GFAP concentration values after 3 h of sevoflurane anesthesia (2257.3 and 1794.0 pg mL^−1^). They received cochlear implants for bilateral hearing loss at birth, and the preoperative social maturity scale showed normal psychological development [32]. Further studies are needed to identify individual variations of susceptibility to neuronal injury during anesthesia induction.

## 5. Conclusions

In conclusion, prolonged sevoflurane anesthesia did not increase the neuronal injury assessed by the plasma biomarker in pediatric patients younger than 3 years old. This is a preliminary study with a small dataset to identify the timely changes of plasma glial fibrillary acidic protein concentration in pediatric patients with sevoflurane anesthesia with or without dexmedetomidine. More research should be carried out to substantiate the findings that were missed in this preliminary study.

## Figures and Tables

**Figure 1 jcm-10-00662-f001:**
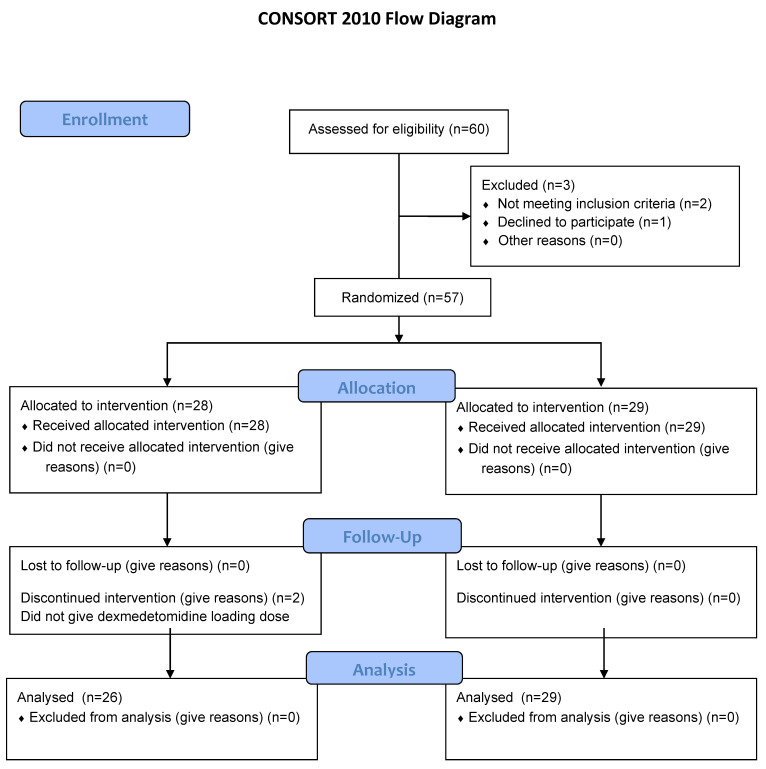
CONSORT diagram.

**Figure 2 jcm-10-00662-f002:**
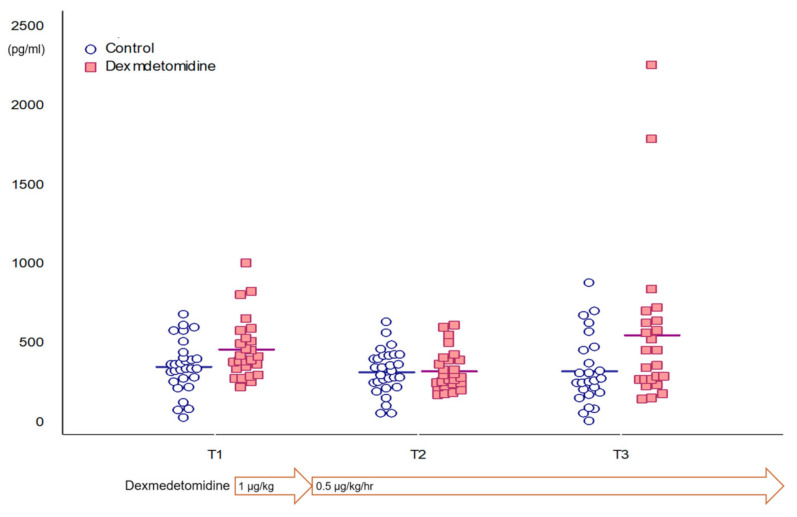
Plasma glial fibrillary acidic protein concentration differences during surgery between the groups.

**Table 1 jcm-10-00662-t001:** Baseline demographic and clinical characteristics for each group.

	Dexmedetomidine Group (n = 26)	Control Group (n = 29)
Age (months)	12.0 (4.8)	13.7 (6.5)
Weight (kg)	9.6 (1.8)	10.1 (1.9)
Height (cm)	76.2 (6.4)	76.0 (8.3)
Gender (male, %)	11 (42)	20 (69)
Total anesthesia time (min.)	285.7 (73.4)	251.1 (69.1)
Type of operation (n)		
Cochlear implantation	19	14
Penoplasty	3	6
Syndactyly division	4	7
Suturectomy	0	2
Fraction of expired sevoflurane (%)	1.3 (0.3)	2.5 (0.6)
End tidal carbon dioxide (mmHg)	37 (2.1)	38 (2.0)
Bispectral index	61.6 (8.4)	61.5 (4.5)
Duration of hypotension (%)	2.1 (3.5)	1.2 (3.0)
Presence of arrhythmia	None	None

Data are n (%), mean (SD, range), and median (interquartile range, IQR), unless stated otherwise.

**Table 2 jcm-10-00662-t002:** Plasma glial fibrillary acidic protein concentration (pg·mL^−1^) during surgery.

	T1	T2	T3	*p*-Value
All patients	387.7 (298.9–510.8)	329.1 (131.5)	312.9 (233.8–576.2)	0.072
Control group	358.1 (162.7)	325.8 (138.3)	332.0 (220.1)	0.759
Dexmedetomidine group	406.7 (342.9–534.3)	332.8 (126.1)	412.4 (273.9–638.8)	0.044

Data are mean (SD), and median (IQR), unless stated otherwise.

## Data Availability

The data presented in this study are available on request from the corresponding author.

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
