# Peer review of "Changes in Plasma Glial Fibrillary Acidic Protein in Children Receiving Sevoflurane Anesthesia: A Preliminary Randomized Trial"

_jcm, 2021, doi:10.3390/jcm10040662_

Round 1
Reviewer 1 Report
Dear Authors,
Congratulations for a good piece of scientific job.
Nevertheless, I should have some comments concerning methodology, because as far as I can see, the authors need some advice concerning BIS-guided general anaesthesia using sevoflurane.
line 79-85
Proper depth of general anaesthesia is usually defined by BIS value of 40-60, whereas the authors defined the depth of general anaesthesia by BIS value between 40-70 - please comment why, because the range is too wide. OR
because the trial is over, that cannot be changed so I would get over the problem with the insertion of the following phrase:
Because the intraoperative vasoactive reaction expressed by heart rate and blood pressure variations are dependent on the depth of anaesthesia, sevoflurane was maintained guided by a bispectral index of 40–70 during the surgery with a target value of around 60, to enable the futher precision of remifentanil dosing. Additionally, as BIS happens to show aberrant values during different stages of general anaesthesia using sevoflurane and rocuronium, therefore the BIS values were on-line verified against observance of Fraction of Inspired Anaesthetic Agent (FiAA), Fraction of Expired Anaesthetic Agent (FeAA) and MAC (Mean Alveolar Concentration).
please cite :
Vakkuri A, Yli-Hankala A, Särkelä M, et al. Sevoflurane mask induction of anaesthesia is associated with epileptiform EEG in children. Acta Anaesthesiol Scand 2001; 45:805–811.
Stasiowski MJ, Marciniak R, Duława A, Krawczyk L, Jałowiecki P. Epileptiform EEG patterns during different techniques of induction of general anaesthesia with sevoflurane and propofol: a randomised trial. Anaesthesiol Intensive Ther. 2019;51(1):21-34. doi: 10.5603/AIT.a2019.0003. Epub 2019 Feb 6. PMID: 30723886.
Tallach RE,Ball DR, Jefferson P. Monitoring seizures with the bispectral index. Anaesthesia 2004; 59: 1033–4.
Yue H, Han J, Liu L, Wang K, Li J. Effect of rocuronium on the bispectral index under anesthesia and tracheal intubation.Exp Ther Med. 2016 Dec;12(6):3785-3789. doi: 10.3892/etm.2016.3829. Epub 2016 Oct 20
Dahaba AA, Mattweber M, Fuchs A, Zenz W, Rehak PH, List WF, Metzler H. The effect of different stages of neuromuscular block on the bispectral index and the bispectral index-XP under remifentanil/propofol anesthesia. Anesth Analg. 2004 Sep;99(3):781-7, table of contents.
If it is possible, I would optionally resign in methodology from Vol% and present FiAA, FeAA, MAC in the table 1, which is the current standard in the contemporary literatures.
Moreover, hyperventialtion is dangerous for children anaesthetised with sevoflurane. I would opt for presenting of mean end-tidal CO2 to be within range of 35-37 during the conduction of general anaesthesia, so that the methodology is more professionally designed. Mean end-tidal CO2 influences cerebral blood flow, especially when children are hyperventilated, what is a fact known from physiology. Some reviewers may find it a huge limitation if the methodology lacks of this data, because it may influence the plasma glial fibrillary acidic protein concentration to a greater extent than exposure to sevoflurane.
Best wishes to Authors
I am looking forward to receiving your revised version of the manuscript.
Should the comments be taken into consideration, I would opt for publication of the manuscript without further changes.
Author Response
Response to Reviewer 1 Comments
Point 1: Congratulations for a good piece of scientific job. Nevertheless, I should have some comments concerning methodology, because as far as I can see, the authors need some advice concerning BIS-guided general anaesthesia using sevoflurane (line 79-85). Proper depth of general anaesthesia is usually defined by BIS value of 40-60, whereas the authors defined the depth of general anaesthesia by BIS value between 40-70 - please comment why, because the range is too wide. OR because the trial is over, that cannot be changed so I would get over the problem with the insertion of the following phrase:
Because the intraoperative vasoactive reaction expressed by heart rate and blood pressure variations are dependent on the depth of anaesthesia, sevoflurane was maintained guided by a bispectral index of 40–70 during the surgery with a target value of around 60, to enable the futher precision of remifentanil dosing. Additionally, as BIS happens to show aberrant values during different stages of general anaesthesia using sevoflurane and rocuronium, therefore the BIS values were on-line verified against observance of Fraction of Inspired Anaesthetic Agent (FiAA), Fraction of Expired Anaesthetic Agent (FeAA) and MAC (Mean Alveolar Concentration).
please cite :
Vakkuri A, Yli-Hankala A, Särkelä M, et al. Sevoflurane mask induction of anaesthesia is associated with epileptiform EEG in children. Acta Anaesthesiol Scand 2001; 45:805–811.
Stasiowski MJ, Marciniak R, Duława A, Krawczyk L, Jałowiecki P. Epileptiform EEG patterns during different techniques of induction of general anaesthesia with sevoflurane and propofol: a randomised trial. Anaesthesiol Intensive Ther. 2019;51(1):21-34. doi: 10.5603/AIT.a2019.0003. Epub 2019 Feb 6. PMID: 30723886.
Tallach RE,Ball DR, Jefferson P. Monitoring seizures with the bispectral index. Anaesthesia 2004; 59: 1033–4.
Yue H, Han J, Liu L, Wang K, Li J. Effect of rocuronium on the bispectral index under anesthesia and tracheal intubation.Exp Ther Med. 2016 Dec;12(6):3785-3789. doi: 10.3892/etm.2016.3829. Epub 2016 Oct 20
Dahaba AA, Mattweber M, Fuchs A, Zenz W, Rehak PH, List WF, Metzler H. The effect of different stages of neuromuscular block on the bispectral index and the bispectral index-XP under remifentanil/propofol anesthesia. Anesth Analg. 2004 Sep;99(3):781-7, table of contents.
Response 1: Thank you for your insightful comments. Your comments were helpful for significantly improving this manuscript.
We added phrase as your suggestion. Thank you for your consideration.
Point 2: If it is possible, I would optionally resign in methodology from Vol% and present FiAA, FeAA, MAC in the table 1, which is the current standard in the contemporary literatures.
Response 2: Thank you for your comments. We revise the table as your recommendation.
Point 3: Moreover, hyperventialtion is dangerous for children anaesthetised with sevoflurane. I would opt for presenting of mean end-tidal CO2 to be within range of 35-37 during the conduction of general anaesthesia, so that the methodology is more professionally designed. Mean end-tidal CO2 influences cerebral blood flow, especially when children are hyperventilated, what is a fact known from physiology. Some reviewers may find it a huge limitation if the methodology lacks of this data, because it may influence the plasma glial fibrillary acidic protein concentration to a greater extent than exposure to sevoflurane.
Response 3: Thank you for your comments. We tried to maintain normocapnia (end tidal CO2 35-40 mmHg) during the surgery (mean EtCO2 37 ± 2.1mmHg for dexmedetomidine group, 38 ± 2.0mmHg for control group). We added the end tidal CO2 in the table as your recommendation.
Reviewer 2 Report
Dear Dr. Kim,
Thank you for allowing me to review your manuscript on the effects of sevoflurane and dexmedetomidine on glial fibrillary acidic protein in a pediatric surgical population under three years of age. You have presented a necessary pilot study looking at potential effects of two anesthetics in conjunction with minor surgery in a pediatric population on a known biomarker of neuronal injury that has previously been associated with traumatic brain injury, glial fibrillary acidic protein. I have a number of concerns with the ability to draw conclusions from the study as presented, some of which you yourself cite within the manuscript as known caveats.
The primary issue, which you touch on in the discussion, is the lack of baseline GFAP for the two groups. This lack of baseline especially makes it difficult to assess the role of dexmedetomidine and to differentiate its effect from the potential GFAP decreasing effect from sevoflurane itself. While dexmedetomidine has been shown in adult surgical geriatric populations to decrease intraoperative GFAP (PMID: 33411362), sevoflurane in and of itself and in the absence of surgery decreases adult GFAP levels (PMID: 32536445.) Given that dexmedetomidine was used intraoperatively as a sevoflurane-sparing adjuvant, with the sevoflurane-only group receiving twice the dose of sevoflurane as the dexmedetomidine group, its completely unclear whether we are seeing a dose-dependent effect of sevoflurane as opposed to any effect at all from dexmedetomidine. What’s more, while there is statistical significance in the GFAP levels from T2 to T3 (which itself appears to be driven entirely by two outliers) it was not stated and is unclear if there is any difference between T1 and T3. If there is not, it seems more likely that the results represent stochastic fluctuations over time rather than any real trend related to treatment. Also given the dexmedetomidine group trends higher than the control group, whether this is due to baseline differences cannot be assessed either.
As the study stands, all that can potentially be concluded is whether there is a trend in GFAP intraoperatively over time with the two regimens. Without a baseline, the regimens should not be compared given the difference in sevoflurane dose as well as dexmedetomidine dose and the unclear effect of each and the effect of surgery itself in a pediatric population. Your conclusions as stated in the conclusions section were correct, but I think there needs to be additional delineation of the limits of possible inference based on the issues with this preliminary study.
Thank you, and I look forward to following your future work in this area.
Author Response
Response to Reviewer 2 Comments
Point 1: Thank you for allowing me to review your manuscript on the effects of sevoflurane and dexmedetomidine on glial fibrillary acidic protein in a pediatric surgical population under three years of age. You have presented a necessary pilot study looking at potential effects of two anesthetics in conjunction with minor surgery in a pediatric population on a known biomarker of neuronal injury that has previously been associated with traumatic brain injury, glial fibrillary acidic protein. I have a number of concerns with the ability to draw conclusions from the study as presented, some of which you yourself cite within the manuscript as known caveats.
The primary issue, which you touch on in the discussion, is the lack of baseline GFAP for the two groups. This lack of baseline especially makes it difficult to assess the role of dexmedetomidine and to differentiate its effect from the potential GFAP decreasing effect from sevoflurane itself. While dexmedetomidine has been shown in adult surgical geriatric populations to decrease intraoperative GFAP (PMID: 33411362), sevoflurane in and of itself and in the absence of surgery decreases adult GFAP levels (PMID: 32536445.) Given that dexmedetomidine was used intraoperatively as a sevoflurane-sparing adjuvant, with the sevoflurane-only group receiving twice the dose of sevoflurane as the dexmedetomidine group, its completely unclear whether we are seeing a dose-dependent effect of sevoflurane as opposed to any effect at all from dexmedetomidine. What’s more, while there is statistical significance in the GFAP levels from T2 to T3 (which itself appears to be driven entirely by two outliers) it was not stated and is unclear if there is any difference between T1 and T3. If there is not, it seems more likely that the results represent stochastic fluctuations over time rather than any real trend related to treatment. Also given the dexmedetomidine group trends higher than the control group, whether this is due to baseline differences cannot be assessed either.
As the study stands, all that can potentially be concluded is whether there is a trend in GFAP intraoperatively over time with the two regimens. Without a baseline, the regimens should not be compared given the difference in sevoflurane dose as well as dexmedetomidine dose and the unclear effect of each and the effect of surgery itself in a pediatric population. Your conclusions as stated in the conclusions section were correct, but I think there needs to be additional delineation of the limits of possible inference based on the issues with this preliminary study.
Thank you, and I look forward to following your future work in this area.
Response 1:
- We thank you for reviewing our manuscript and also for your insightful comments and questions.
- The plasma GFAP concentration was not differ between T1 and T3 (P=0.473) in dexmedetomidine group. We added this in the results.
- We agree with your opinion on this preliminary study. We could get more meaningful results through preoperative GFAP measurements. We added phrase in the conclusion about preliminary design in more detail.
à This is a preliminary study with a small data set to identify the timely changes of plasma glial fibrillary acidic protein concentration in pediatric patients with sevoflurane anesthesia with or without dexmedetomidine. More research should be carried out to substantiate these findings which were missed in this preliminary study.
Round 2
Reviewer 2 Report
Thank you for your revisions, I look forward to the follow-up study on this.